# Modified Vaginal Mesh Procedure with DynaMesh^®^-PR4 for the Treatment of Anterior/Apical Vaginal Prolapse

**DOI:** 10.3390/diagnostics13182991

**Published:** 2023-09-18

**Authors:** Chia-Ju Lin, Chih-Ku Liu, Hsiao-Yun Hsieh, Ming-Jer Chen, Ching-Pei Tsai

**Affiliations:** 1Department of Obstetrics and Gynecology, Taichung Veterans General Hospital, Taichung 40705, Taiwan; cjl0401@vghtc.gov.tw (C.-J.L.); k388@vghtc.gov.tw (M.-J.C.); 2Ph.D. Program in Translational Medicine, National Chung Hsing University, Taichung 40204, Taiwan; 3Rong Hsing Research Center for Translational Medicine, National Chung Hsing University, Taichung 40204, Taiwan

**Keywords:** female, pelvic organ prolapse, surgical mesh, uterine prolapse, vagina, cystocele, prostheses and implants

## Abstract

(1) Background: Treating female pelvic organ prolapse (POP) is challenging. Surgical meshes have been used in transvaginal surgeries since the 1990s, but complications such as mesh exposure and infection have been reported. Polyvinylidene fluoride (PVDF) mesh, known for its stability and non-reactive properties, has shown promise in urogynecological surgeries. (2) Methods: A retrospective analysis was conducted on 27 patients who underwent a modified PVDF vaginal mesh repair procedure using DynaMesh^®^-PR4 and combined trans-obturator and sacrospinous fixation techniques. Additional surgeries were performed as needed. (3) Results: The mean operation time was 56.7 min, and the mean blood loss was 66.7 mL. The average hospitalization period was 4.2 days with Foley catheter removal after 2 days. Patients experienced lower pain scores from the day of the operation to the following day. Postoperative follow-up revealed that 85.2% of patients achieved anatomic success, with 14.8% experiencing recurrent stage II cystocele. No recurrence of apical prolapse was observed. Complications were rare, with one case (3.7%) of asymptomatic mesh protrusion. (4) Conclusions: The modified vaginal mesh procedure using DynaMesh^®^-PR4 showed favorable outcomes with a short operation time, low recurrence rate, rare complications, and improved functional outcomes. This surgical option could be considered for anterior and apical pelvic organ prolapse in women.

## 1. Introduction

The treatment of female pelvic organ prolapse (POP) poses a significant challenge. Advanced stages of POP are associated with somatic symptoms such as voiding dysfunction and obstructive defecation [1]. In order to alleviate these symptoms and enhance the patient’s quality of life, an ideal treatment approach should aim to restore the anatomical defect [2] and minimize the need for repeat operations.

In the 1990s, surgical meshes were developed for use in transvaginal surgeries for POP and stress urinary incontinence (SUI) [3,4] following their initial application in hernia repair. While transvaginal mesh (TVM) procedures have shown better surgical outcomes compared to native tissue operations [5,6,7], there are still some side effects that have been reported. Previous studies have identified complications such as mesh exposure, mesh contraction [8], infection at the site of mesh exposure, dyspareunia, and voiding dysfunction [9].

In 2008, the United States Food and Drug Administration (FDA) issued a safety warning and released the first report of transvaginal mesh procedure [10]. A decade later, the FDA announced a prohibition on the sale of anterior transvaginal mesh implants in the United States [10]. However, in certain Asian countries, gynecological/urogynecological specialists continue to utilize mesh more frequently than native tissue for pelvic reconstruction surgery in stages III and IV of POP [4], with complications being less frequently reported. The key considerations include using an appropriate mesh material [11] and specialized training in surgical techniques [4].

Polyvinylidene fluoride (PVDF) is a thermoplastic fluoropolymer known for its good biostability and non-reactive properties [12,13]. In comparison to polypropylene (PP) mesh, PVDF mesh exhibits lower stiffness, resulting in less fibrosis [13], foreign body reaction, and scar formation in the surrounding tissue [12]. Moreover, it provides enhanced stability of the structural position [14]. In 2017, Barski et al. conducted the first cohort study investigating the use of PVDF in urogynecological surgeries [9]. Similarly, Eslami et al. demonstrated the safety of PVDF in anterior vaginal wall repair and the trans-obturator technique (TOT) [11]. DynaMesh^®^ PR4 is the only PVDF monofilament mesh implant available worldwide for cystocele repair. The objective of our study was to investigate the safety and efficacy of transvaginal POP repair using DynaMesh^®^-PR4, employing a modified technique involving combined trans-obturator and sacrospinous fixation.

## 2. Materials and Methods

This retrospective, single-centered study was conducted at the Taichung Veterans General Hospital, Taiwan. The study received approval from the institutional review board (Approval number: CE22272B). Clinical data were collected from 2020 until the end of 2021.

### 2.1. Study Population

The study enrolled patients with symptomatic pelvic organ prolapse, specifically stage II to III, who underwent transvaginal pelvic reconstruction surgery using PVDF (DynaMesh^®^ PR4) mesh. The medical records of twenty-seven patients who underwent the modified vaginal mesh repair procedure were retrospectively reviewed. The patients were informed about the benefits and potential complications associated with the surgical procedure involving synthetic meshes, and their consent was obtained as part of routine practice. Exclusion criteria included a history of previous vaginal mesh surgery, plans for pregnancy, a history of cancer with chemotherapy or radiotherapy, cervical dysplasia history, or infection at the vaginal mucosa. Some patients presented with anterior or posterior wall prolapse, elongation of the cervical stump, or stress urinary incontinence (SUI).

### 2.2. Surgical Technique and Operative Care

In contrast to the original surgical procedure involving DynaMesh-PR4 mesh, which utilized four arms of the mesh extending from the incision in the anterior vaginal wall to the obturator foramen, we modified the technique by altering the method of posterior arm penetration to enhance the strength of the apical suspension.

The operation comprised four major steps: (Figure 1):Step 1: Vaginal Dissection

A 5 cm-long midline incision was made on the anterior vaginal wall, approximately 2 cm below the mid-urethra, extending toward the cervix or vaginal cuff. Dissection was performed using sharp or blunt techniques, exposing the vesicovaginal septum anteriorly, the paravaginal space and the pubic rami laterally, the arcus tendinous of the endopelvic fascia parallelly, and the edge of the cervical ring posteriorly.

Step 2: Insertion of Distal Arms

The paravaginal space was dissected toward the bilateral ischiorectal fossa using scissors and blunt dissection. Two 2 cm-long skin incisions were made bilaterally at the level of the middle of the clitoris and urethral orifice, which is known as the genitofemoral folds. A transobturator trocar was inserted through the incision, penetrating the obturator muscles and rotating anteromedially toward the dissected space beside the bladder neck under finger-guided tactile sensation. The anterior arms of the DynaMesh-PR4 mesh were tied to the tip of the trocar and pulled out bilaterally from the incisions.

Step 3: Insertion of Proximal Arms

The incisions for trocar guidance for the posterior arms were marked on the buttock, 3 cm lateral to the anus and 3 cm inferiorly bilaterally. The ischial spine was identified as a landmark. The trocar was inserted through the buttock incisions, penetrating the sacrospinous ligament and reaching the paravaginal dissected space. The posterior arms of the DynaMesh-PR4 mesh were tied to the trocar tip and pulled out bilaterally through the sacrospinous ligament (Figure 2).

Step 4: Mesh Fixation and Vaginal Closure

After extracting the four arms of the mesh, the position of the mesh was adjusted to attach to the anterior vaginal wall and elevate the prolapse to the level of sacrospinous ligament. Once the Dynamesh PR4 mesh was fully applied, it was sutured to the anterior vaginal wall and cervix using 2-0 polydioxanone sutures for fixation. If necessary, the vaginal mucosa was trimmed, and closure was performed using 2-0 polyglactin sutures in a continuous, double-layer method.

If the patient needed a concurrent mid-urethral sling for stress incontinence, the procedure was performed separately following the prolapse surgery. Patients with cystocele, enterocele, or rectocele underwent anterior vaginal wall repair before the mesh procedure, and perineorrhaphy was performed after the mesh procedure. In cases where the patient had an elongated cervical stump, a trachelectomy was conducted. Cystoscopy was performed to rule out bladder or urethral perforation. As a prophylactic measure, patients were prescribed 1 g of intravenous cefazolin before the operation and oral cefadroxil (500 mg, twice daily) for postoperative antibiotics spanning 7 days. Topical estrogen cream or estrogen supplements were not routinely administered following the surgery. Vaginal packing was removed within 24 h postoperatively, while urethral catheters were left in place for 48 h. The discharge criteria included two instances of adequate post-void residual urine (i.e., <100 mL) and the patient’s overall good general condition. Follow-up outpatient visits were scheduled at 1 week, 1 month, 3 months, and yearly intervals after discharge.

### 2.3. Assessment and Analysis

We assessed the pelvic organ prolapse symptoms using the Pelvic Floor Disability Index (PFDI-20) [15] and the Pelvic Organ Prolapse Quantification (POP-Q) system [16]. During outpatient visits, patients underwent pelvic examination in the lithotomy position conducted by experienced gynecologists to evaluate the position and stage of prolapse. Additional tests such as pap smears, urinary analysis, and pelvic ultrasonography were conducted to exclude inflammation and malignancy. Preoperatively, urodynamic studies (UDS) and pad tests were performed to evaluate incontinence symptoms. A urodynamic study (UDS) and pad test were performed preoperatively. The symptoms of incontinence were evaluated using the Incontinence Impact Questionnaire, Short Form (IIQ-7). Sexual function was assessed using the Pelvic Organ Prolapse/Urinary Incontinence Sexual Questionnaire (PISQ-12) [17]. Postoperative UDS was not routinely performed.

The parameters, including operative time, blood loss, visual analog scale (VAS) scores on the day of operation and first postoperative day, Foley catherter removal time and hospitalization duration, were analyzed. After discharge, outpatient visits were scheduled for one week and four weeks later to check the complications and complete questionnaires. The failure of operation or recurrent prolapse was defined as the absence of anterior vaginal wall at POP-Q stage II (Aa and Ba between −1 to +1cm). Yearly outcomes were reviewed and recorded at the subsequent outpatient department visits.

Continuous variables were presented as mean ± standard deviation, while categorical variables were presented as numbers and percentages. We used the Wilcoxon signed rank test and McNemar’s test to compare the data before and after the surgery. All analyses were conducted using SPSS software version 24 (SPSS, Chicago, IL, USA). In all analyses, a value of *p* < 0.05 was considered statistically significant.

## 3. Results

Twenty-seven patients with anterior vaginal wall prolapse who underwent modified PVDF vaginal mesh surgery were included in the analysis. The basic characteristics and perioperative information are listed in Table 1. The mean age of the patients was 66 ± 5.5 years old. The mean parity was 3.3 ± 1.2, mostly vaginal deliveries, and the mean body max index (BMI) was 25.3 ± 3.2. Five patients had undergone hysterectomy previously. Before the operation, two patients (7.4%) had the pelvic prolapse POP-Q stage II, twenty-three patients (85.2%) were at POP-Q stage III and the other two patients (7.4%) were at stage IV.

The mean operation time for the modified PVDF vaginal mesh surgery exclusively was 56.7 ± 13.4 min. Concomitant surgeries are detailed in Table 1. The most commonly performed concurrent procedures were perineorrhaphy (24/27, 88.90%) and VPUS (16/27, 59.30%), which were both utilized for level III support. Other concurrent procedures were less frequent, such as trachelectomy (3/27, 11.1%) and laparoscopic salpingo-oophorectomy (1/27, 3.7%). The mean blood loss was 66.7 ± 36.7 mL, and no patient required a blood transfusion. The mean hospitalization period was 4.2 ± 0.9 days, and the Foley catheter was removed on average after 2 days. The mean visual pain score on the day of the operation was 3.0 ± 1.6, which decreased to 1.9 ± 0.9 on the following day (Table 1).

The preoperative and postoperative POP-Q reference points are shown in Table 2. The mean follow-up period was 3 years. Twenty-three patients (85.2%) achieved postoperative anatomic success, while four patients (14.8%) experienced recurrent stage II cystocele (Table 3). All recurrent cases were asymptomatic and limited to the hymen area. No recurrence of apical prolapse was observed. The PFDI-20 and IIQ-7 scores significantly improved at the 3 year follow-up (*p* < 0.0001) (Table 2). Only four patients were sexually active in our study, and they reported progressive improvement in sexual function scores (PISQ-12) after the operation.

Other postoperative conditions are listed in Table 3. During outpatient follow-up, one patient (3.7%) was found to have mesh protrusion. No vaginal oozing, pain, or negative impact on intercourse was reported. One patient (3.7%) complained of buttock pain during a visit. Two patients (7.4%) experienced persistent urinary incontinence even after concomitant VPUS surgery, and one patient developed de novo stress urinary incontinence at the 3-year follow-up. No pelvic hematoma or inflammatory disease were reported.

## 4. Discussion

This study showed that modified PVDF vaginal mesh surgery with DynaMesh-PR4 mesh is an effective method for apical–anterior vaginal prolapse suspension. The three-year objective outcome revealed a subjective/objective success rate of 100%/85.2%, respectively. The recurrent rate was 14.8%, with all cases involving in the anterior compartment, or within the hymen, and all were asymptomatic. Overall, there were improvements in vaginal symptoms, quality of life, and sexual satisfaction postoperatively, as measured by PFDI-20, IIQ-7, and PISQ-12.

DynaMesh^®^-PR4 was initially designed for cystocele repair, with four arms inserted using the double trans-obturator technique (TOT). A trocar is inserted through the foramen obturatorium, over the insertion of the arcus tendinous, but without direct penetration of the sacrospinous ligament [9,11]. Therefore, this technique alone cannot provide adequate apical support, and some patients still require concomitant apical fixation, such as sacrospinal fixation or sacrocolpopexy [9]. In our study, we modified the proximal trocar fixation point to achieve the effect of apical suspension by making buttock incisions and penetrating the sacrospinous ligament. Postoperatively, the baseline C point of the POP-Q score ranged from 0 to –5 cm (*p* < 0.001), and the D point ranged from −3 to −4.75 cm (*p* < 0.001). The modified procedure not only provides level II support but also provides level I suspension support to the cervix or vaginal cuff. Currently, many vaginal mesh products are designed to treat anterior/apical prolapse concurrently [18,19,20]. Chen et al. concluded that if surgery is performed concomitantly to support the apical and anterior compartment, the risk of recurrence decreases significantly [21]. In this trial, no cases of relapsed apical prolapse were found, which indicates the importance of the adequate apical fixation over the sacrospinous ligament.

However, recurrence of the cystocele is the toughest issue in prolapse repair. In our trial, there were four recurrent cases (4/27, 14.8%), all of which occurred in the anterior compartment. Fortunately, none of the patients experienced symptoms (within hymen), and none required re-operation. The recurrence rate in this study is comparable with our previous studies of vaginal meshes (from the same institution and performed by the same surgical team), which showed a failure rate of 13 to 20% [22,23,24]. Recently, it has been suggested that anatomic definitions are too strict, as more than 75% of women presenting for annual examinations without POP symptoms would not meet the criteria for “optimal anatomic outcome” [25]. Some studies have defined Ba point <0 as anatomical success, since patients usually have no symptoms in this situation [26,27].

Follow-up evaluation revealed favorable functional outcomes were achieved. Various pelvic symptoms and their negative impacts on quality of life improved significantly. No serious complications were noted in this study. Only one case (3.7%) reported postoperative pain in the pelvic floor. However, this patient also underwent posterior colporrhaphy concurrently, which may have contributed to buttock pain. Postoperative pain is not necessarily caused by the mesh itself. Pain is one of the most concerning complications of TVM surgery [18]. Sanses et al. conducted a study involving 206 cases of vaginal mesh with a similar procedure, which showed that 6.8% of the cases developed buttock pain and 2.4% experienced defecatory pain. Compared with traditional PP meshes, PVDF meshes demonstrated better biostability, less focal inflammation or fibrosis in host tissue, and lower bending stiffness in a previous study [13]. Scarring resulting from the mesh is thought to play a role in the development of pain, as scar contraction leads to a decreased elasticity and a stiff mesh/tissue compound. Therefore, it appears that PVDF meshes produce less scarring due to their higher biocompatibility, reducing the incidence of postoperative chronic pain complications.

Only one case (1/27, 3.7%) reported mesh exposure during the follow-up in our study. She did not experience any symptoms. A tiny mesh erosion was noticed during the follow-up visit. The protruding mesh was directly trimmed, and Premarin vaginal cream was prescribed. No sequelae occurred thereafter. In previous studies using PVDF mesh, the mesh erosion rate was about 5 to 6% [9,11] compared to 10–19% with PP mesh [8,18,28]. The reduction in mesh erosion observed in our study may be attributed to the double-layer closure method, tension-free mesh fixation, and the advantages of using PVDF material. Given that the FDA has determined that serious adverse events associated with previous vaginal meshes are not rare [10], it is essential to report further results regarding different new mesh products.

De novo SUI can be a concern after vaginal mesh surgeries. Previous studies have reported a range of 9.9% to 1.2% for the risk of a significant decrease in maximum urethral closure pressure [29,30,31]. The risk of de novo SUI after a modified Surelift (PP mesh) procedure was shown to be significantly increased at 18.9% [19]. In contrast, we observed two cases (7.4%) that developed mild de novo SUI after DynaMesh^®^—PR4 surgery. None of these cases required re-operation for incontinence. Careful paravesical dissection and reduced tissue trauma/inflammation after PVDF mesh surgery are the key factors contributing to this success.

There were several limitations in our study. First, it was a single-arm design study, lacking a comparison with operations using vaginal mesh made of other materials or native tissue repair of pelvic organ prolapse. Second, although we reported high success rates, with high satisfaction and low complications, our sample size was relatively small. Nonetheless, the outcomes and success rates are consistent with those observed in our historic cohort of patients who underwent surgery using other techniques for POP repair [22,23,24]. Additionally, the population in our study was predominantly sexually inactive. While the average follow-up PISQ 12 score was higher, only 15% (4/27) of the patients completed the questionnaire. Others reported that they were not sexually active and rarely engaged in intercourse. Thus, cultural differences may have influenced the surgical outcomes. Further studies involving sexually active subjects are needed to draw conclusions regarding sexual symptoms. The strengths of our study included a homogeneous population of patients, a standardized evaluation protocol and a mid-term follow-up period.

## 5. Conclusions

Considering the existing evidence highlighting serious and well-recognized safety concerns with transvaginal mesh repair, there is an urgent need to explore alternative options. With a short operation time, low recurrence rate, rare complications, and good functional outcomes, pelvic floor reconstructive surgery using the modified vaginal mesh procedure (DynaMesh^®^—PR4) could be considered as a surgical option for anterior and apical pelvic organ prolapse in women.

## Figures and Tables

**Figure 1 diagnostics-13-02991-f001:**
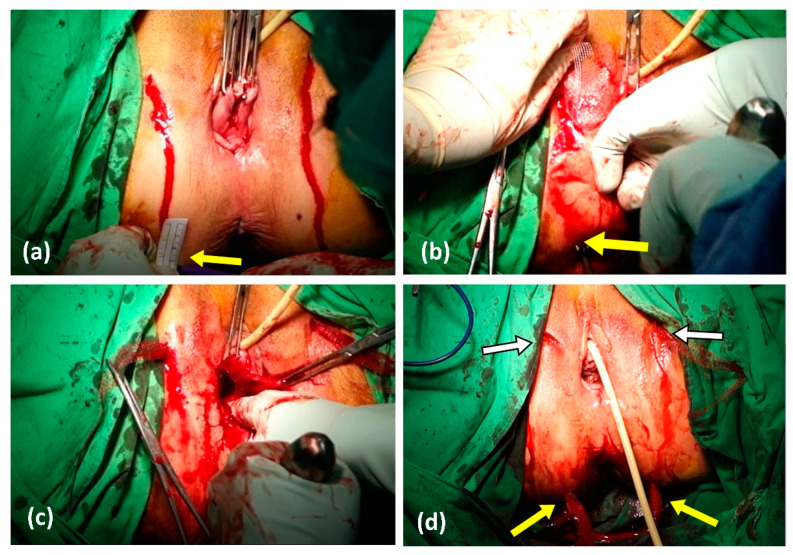
Steps of operation. (**a**) We marked the buttock site where the guiding trocar was planned to be applied: horizontally, 3 cm laterally from the anus and 3 cm vertically below those points. (**b**,**c**) After the anterior arms of DynaMesh-PR4 mesh were applied outside-in through bilateral obturator foramen, the posterior arms were applied from the anterior vaginal incision, guided by the trocar. The trocar was inserted from the marked spot, passed through the sacrospinous ligament, into the vaginal incision. (**d**) The wounds of the four arms of meshes are indicated by the yellow arrow for the skin incision of the posterior arms of the DynaMesh-PR4 mesh and by the white arrow with a black outline for the skin incision of the anterior arms of the DynaMesh-PR4 mesh.

**Figure 2 diagnostics-13-02991-f002:**
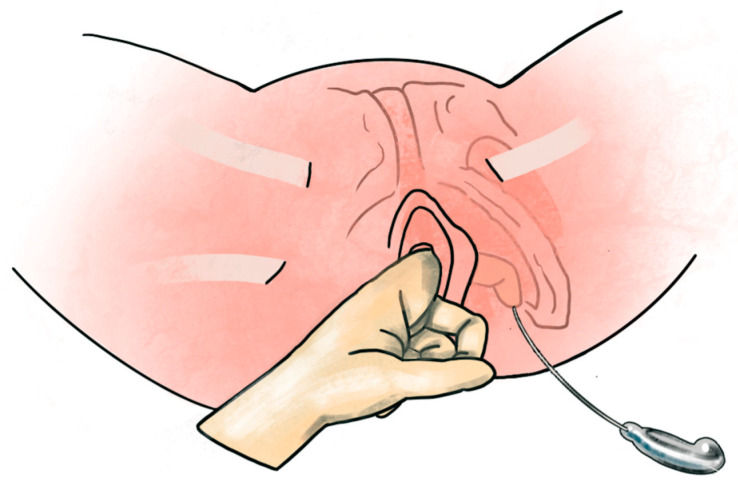
Step 3 of operation. Under the guidance of the index finger of the operator, the trocar was inserted from the marked spot, passed through the sacrospinous ligament, and into the vaginal incision. The posterior arms were applied from the anterior vaginal incision, guided by the trocar and pulled out at the skin incision.

**Table 1 diagnostics-13-02991-t001:** Preoperative characteristics of patients (*n* = 27).

Basic Data	Total (*n* = 27)	Range
Mean age (range)	66.3 ± 5.5	53–72
Median parity (range)	3	0–6
Menopause	100%	27/27
BMI (Kg/M2) (range)	25.3 ± 3.2	18.8–30.7
Sexually active (%)	7/27	25.90%
Prior hysterectomy (%)	5/27	18.50%
POP-Q		
Stage I	0	0%
Stage II	2/27	7.40%
Stage III	23/27	85.20%
Stage IV	2/27	7.40%
Perioperative data		
Mean operation time (min)	56.7 ± 13.4	30–85
Blood loss (mL)	66.7 ± 36.7	50–200
Mean pain score (VAS) Operative day	3.7 ± 1.6	1–7
Postoperative day 1	1.9 ± 0.9	1–3
Mean hospitalized day	4.2 ± 0.9	3–6
Mean Foley day	2.0 ± 0	2–5
Concomitant surgeries		
VPUS	16/27	59.30%
Perineorrhaphy	24/27	88.90%
Trachelectomy	3/27	11.10%
Laparoscopic Salpingo-oophorectomy	1/27	3.70%

POP-Q: Pelvic Organ Prolapse Quantification. VAS: visual analogue scale.

**Table 2 diagnostics-13-02991-t002:** Preoperative POP-Q points of reference at baseline and the questionnaire scores and 3 year follow-up.

	*n*	Preoperative			Postoperative (3 Year)			*p*-Value
Aa	27	0.00	(0.00,	0.50)	−3.00	(−3.00,	−2.50)	<0.001 **
Ba	27	2.00	(1.00,	3.00)	−2.50	(−3.00,	−2.50)	<0.001 **
C	27	0.00	(−1.50,	1.00)	−5.00	(−5.50,	−5.00)	<0.001 **
Gh	27	4.50	(4.00,	5.00)	3.00	(3.00,	3.50)	<0.001 **
Pb	27	3.00	(2.50,	3.00)	4.00	(4.00,	4.50)	<0.001 **
TVL	27	7.00	(7.00,	7.70)	7.00	(7.00,	8.00)	0.034 *
Ap	27	−1.00	(−1.00,	0.00)	−2.00	(−2.00,	−2.00)	<0.001 **
Bp	27	−1.00	(−1.00,	1.00)	−2.00	(−2.00,	−2.00)	<0.001 **
D	22	−3.00	(−4.50,	−1.00)	−4.75	(−5.00,	−4.00)	<0.001 **
PFDI-20	22	41.50	(35.75,	53.25)	23.00	(20.00,	26.00)	<0.001 **
IIQ-7	21	16.00	(12.00,	21.00)	7.00	(7.00,	10.00)	<0.001 **
PISQ 12	4	24.00	(17.25,	27.00)	30.50	(26.75,	34.25)	0.068

PFDI: Pelvic Floor Disability Index. IIQ: Incontinence Impact Questionnaire. PISQ: Pelvic Organ Prolapse/Urinary Incontinence Sexual Questionnaire. Wilcoxon signed rank test, Median (IQR) * *p* < 0.05, ** *p* < 0.01.

**Table 3 diagnostics-13-02991-t003:** Surgical results after 3 years follow-up.

Symptoms	Total (*n* = 27)	
Prolapse Recurrence		
POP-Q Stage II	4	(14.8%)
Stage III	0	(0%)
Stage IV	0	(0%)
Follow-up complications		
Pelvic hematoma	0	(0%)
Pelvic inflammatory disease	0	(0%)
Buttock pain	1	(3.7%)
Constipation	2	(7.4%)
Delayed free voiding (>7 days)	0	(0%)
Persist urine incontinence	2	(7.4%)
De novo stress incontinence	1	(3.7%)
De novo urge incontinence	0	(0%)
Mesh protrusion	1	(3.7%)

POP-Q: Pelvic Organ Prolapse Quantification.

## Data Availability

The data presented in this study are available on request after obtaining additional permission of the Institutional Review Board of Taichung Veterans General Hospital.

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
