# Peer review of "Modified Vaginal Mesh Procedure with DynaMesh^®^-PR4 for the Treatment of Anterior/Apical Vaginal Prolapse"

_diagnostics, 2023, doi:10.3390/diagnostics13182991_

Round 1

Reviewer 1 Report

Dear authors,

it was a pleasure reviewing your manuscript. Female pelvic organ prolapse is a common problem, especially in elderly population, and it`s treatment is a challenge both to gynecologists and urologists. Surgical meshes application is questionable in many centers, so your work may be of interest. 

Study design is good, although your sample is rather small and you lack a control group, which you have commented in Discussion. I think that your study would be of greater importance if you could include a control group.

In the tables legend there should be abbreviations mentioned in the tables. 

Author Response

Dear Reviewer, 

Thank you for your insightful feedback. While we acknowledge that the sample size in our study may be relatively small, the outcomes and recurrence rates observed are consistent with those reported in a previous study conducted by our institution and performed by the same surgical team. Second, we have addressed this point in the discussion section of our manuscript. Besides, we added the abrasions at the end of the tables. We are very grateful for all the points and willing to make adjustments.

Reviewer 2 Report

The manuscript presents first result of using transvaginal mesh from alternative material as in the past. It is prepared as a ''clinical audit'' report. As authors stated the number of included patients is low and there is no comparison with other techniques. This is a major drawback of the study. Even during the discussion authors present a very limited results from other techniques and therefore do not perform almost any comparison.

Also, authors do not comment the fact that almost 90 % of patiens (the great majority) had concomitant perineorrhaphy. It is therefore almost impossible to evaluate the contribution of mesh and contribution of concomitant surgery to the final result. 

Authors could at least include historic cohort of patients with similar clinical characteristics who had other techniques for POP repair (from the same institution and performed by the same surgical team). This would definitively add important and interesting data.

Author Response

Dear Reviewer,

We sincerely appreciate your thoughtful feedback. We acknowledge the limitations of our study, notably the relatively low number of patients and the absence of comparisons with alternative techniques. However, it is crucial to emphasize that our research represents the initial outcomes of utilizing transvaginal PR4 mesh made from alternative materials, with a midterm follow-up period. Furthermore, we added some discussion with our historical data, originating from the same institution and executed by the same surgical team, indicates a comparable recurrence rate, as discussed in our manuscript.

It's worth noting that our approach involves perineorrhaphy for level III defects, while vaginal mesh is predominantly employed for level I-II support. We routinely consider concomitant perineorrhaphy when it is deemed necessary.

Your feedback has been invaluable in refining our study, and we are grateful for your time and attention.

Reviewer 3 Report

The original article with the identifier diagnostics-2531713 represents an interesting approach in treating pelvic organ prolapse using a modified PVDF vaginal mesh repair procedure using DynaMesh® -PR and combined trans-obturator and sacrospinous fixation techniques. 

However, there are several limitations which are accurately addressed in the Discussion section. Retrospective, single-center, single-arm study is very prone to bias and the results of this study cannot be generalized. The biggest pitfall is certainly lack of comparison group and small sample size. 

Introduction section is well written, Methods and Results are clearly presented but, due to the above mentioned limitations, I believe that this study is not quite enough for high-impact journals, such as Diagnostics. 

Author Response

Dear Reviewer,

We are very grateful for the vulnerable feedback. We realize that the sample size in our study may be relatively small. Thus, we have modified the discussion part, and added the comparison of the outcomes and recurrence rates to the previous studies which conducted in the same institution and performed by the same surgical team. The outcome and recurrent rate of our current study is similar to the previous data of different brand of vaginal meshes. We are very grateful for all the points and wiling to make further adjustments.

Round 2

Reviewer 2 Report

I understand that this new technique is similar in outcome as previous techniques from the same centre. What would than be a reason to promote the new technique? Less complications? It is not clear from the manuscript. Also, as almost all patients had additional procedures, next to the mesh, it is impossible to evaluate whether the results are associated with new mesh and technique or other concomitant surgery. The authors did not explain this.

Author Response

Dear Reviewer: 

Thank you for all your precious opinions. Our responses were listed as below: 
1.   As we mentioned in discussion and conclusion paragraphs, pelvic floor reconstructive surgery using the modified PVDF vaginal mesh has the same success rate as traditional PP meh. Besides, this procedure has favorable functional outcomes and fewer complications. Only one case (3.7%) had asymptomatic mesh protrusion, which was lower compared to traditional PP mesh ( 10-19%, reference 8, 18, 27]. No major complication was reported. One patient (3.7%) complained of buttock pain and two patients (7.4%) experienced constipation. 
2.   Since current evidence on the safety of transvaginal mesh repair shows there are serious but well recognized safety concerns, alternatives are urgently needed. More result about different meshes deserve to be reported. PVDF vaginal mesh could be considered as a surgical option for anterior and apical pelvic organ prolapse in women. We had clarify this in the new version.  
3.   In this study, the most frequent performed concomitantly surgical procedures are perineorrhaphy (24/27, 88.90%) and VPUS (16/27, 59.30%). Perineorrhaphy and VPUS are used for level III support, while vaginal mesh is used for level I and II repair. Other concomitant procedures are rare. Therefore, the surgical outcomes of apical and anterior vaginal wall prolapse are mainly rely on PVDF vaginal mesh. This will be added in result section.

In addition to the comments, all spelling and grammatical problems have been checked and corrected.
We look forward to hearing from you in due time regarding our submission and to respond to any further  comments you may have.
Sincerely,

Chai Ju Lin

Reviewer 3 Report

Dear Editor, 

the authors have made the substantial changes according to previous comments - now this manuscript can be included for potential publication in Diagnostics. 

Introduction and Discussion section are nicely presented, Results are clarified and Methodology is quite detailed. 

Minor editing of English language required

Author Response

Dear Reviewer,

Thank you for all the previous opinions. 

In addition to the comments, all spelling and grammatical problems have been checked and corrected.
We look forward to hearing from you soon about our submission and to respond to any further comments you may have.
Sincerely,

Chai Ju Lin

Round 3

Reviewer 2 Report

Thank you very much for responding to comments.

The language is better understandable.